# Whey—A Valuable Technological Resource for the Production of New Functional Products with Added Health-Promoting Properties

**DOI:** 10.3390/foods14244258

**Published:** 2025-12-10

**Authors:** Ewa Czarniecka-Skubina, Marlena Pielak, Katarzyna Neffe-Skocińska, Katarzyna Kajak-Siemaszko, Sabina Karp-Paździerska, Artur Głuchowski, Małgorzata Moczkowska-Wyrwisz, Elżbieta Rosiak, Jarosława Rutkowska, Agata Antoniewska-Krzeska, Dorota Zielińska

**Affiliations:** Department of Food Gastronomy and Food Hygiene, Institute of Human Nutrition Sciences, Warsaw University of Life Sciences (WULS-SGGW), 159C Nowoursynowska St., 02-776 Warsaw, Poland; marlena_pielak@sggw.edu.pl (M.P.); katarzyna_neffe-skocinska@sggw.edu.pl (K.N.-S.); katarzyna_kajak-siemaszko@sggw.edu.pl (K.K.-S.); sabina_karp-pazdzierska@sggw.edu.pl (S.K.-P.); artur_gluchowski@sggw.edu.pl (A.G.); malgorzata_moczkowska@sggw.edu.pl (M.M.-W.); elzbieta_rosiak@sggw.edu.pl (E.R.); jaroslawa_rutkowska@sggw.edu.pl (J.R.); dorota_zielinska@sggw.edu.pl (D.Z.)

**Keywords:** whey, functional products, valorization, sustainable food production

## Abstract

Whey, a by-product of cheese and casein manufacture, represents a major output in dairy processing and a valuable resource for the production of functional foods. This review examines the technological, environmental, and nutritional aspects of whey valorization, emphasizing its transformation from an ecological burden to a raw material with high economic potential. Over time, whey has evolved from being regarded as waste product to becoming a strategic ingredient in the formulation of modern functional foods and bio-based materials. Data from January 2015 to October 2025 were collected from PubMed, Web of Science, and Scopus to outline global whey production, utilization rates, and emerging processing methods. Modern membrane, enzymatic, and non-thermal technologies enable the recovery of valuable components, including proteins, lactose, and bioactive compounds. The use of these techniques reduces the biochemical and chemical oxygen demand in wastewater The review highlights the use of whey in functional beverages, milk and meat processing, edible films, bioplastics, and biofuels, as well as its microbiological and biotechnological potential. Results indicate that only about half of the 180–200 million tonnes of whey produced annually is effectively valorized, underscoring the need for integrated circular-economy approaches. Overall, whey valorization contributes to sustainable food production, environmental protection, and the development of innovative, health-promoting products that align with global strategies for waste reduction and the development of functional foods.

## 1. Introduction

Whey is a liquid by-product remaining after milk coagulation during cheese or casein manufacture. Coagulation primarily occurs as a result of the action of rennet-type enzymes. Whey production accounts for 65–80% of the volume of processed milk [1]. Depending on the pH, which is influenced by the method of milk curd production, the following types of whey are distinguished:-Sweet whey (rennet whey), obtained in the production of hard, semi-hard, and soft cheeses, as well as rennet casein (pH 5.9–6.3).-Acid whey, produced during the cottage cheese production. Coagulation is achieved mainly by acidification (pH 4.3–4.6) [2].

According to some sources, a third type of whey, known as casein whey, is obtained during the casein production process. Strong mineral acids, including sulfuric and hydrochloric acids, are used to precipitate the curd (pH 4.6–4.7) [3].

Whey is rich in many minerals and vitamins (vitamin C, B vitamins, riboflavin—B_2_, and pyridoxine—B_6_), and contains proteins (lactoglobulin, lactalbumin, immunoglobulin, proteose-peptones, lactoferrin), lactose, and trace amounts of fat. The composition of whey varies depending on the season, the source of milk, the type of cheese or yogurt being processed, the technological process, and the method of milk storage after milking. Depending on the processing technology, 50–60% of the milk dry matter ingredients pass into whey, including approximately: 95% of albumin, 95% of globulin, up to 33% of casein (including some coagulum), 96% of lactose, 8% of fat, and 81% of minerals [4].

Due to the lack of sustainable practices, whey is considered a significant source of environmental pollution by the dairy industry, with a high chemical oxygen demand (COD) of 60–80 g/L and a biochemical oxygen demand (BOD) of 30–50 g/L. Large amounts of whey are discharged into wastewater, posing a serious environmental risk. It should be emphasized that significant amounts of whey are not fully utilized and therefore still constitute an ecological problem. Acid whey requires pre-treatment before being discharged into the sewage system, preferably through a combination of physical and chemical processes [2]. The introduction of whey into the environment causes changes in soil structure, reduces crop yields, and, in water bodies, causes the destruction of aquatic ecosystems through the loss of dissolved oxygen [5,6].

On the other hand, milk processing offers significant opportunities for implementing the circular economy model through its use as a feed and energy source, an organic fertilizer [7], a food component, as well as for the production of bioplastics [8], biochemicals [9], and biocatalysts [10]. Therefore, whey can be considered either a waste product or a by-product [11]. By 2030, whey is expected to become an increasingly important strategic ingredient, as growing demand for specialty proteins collides with sustainability-driven supply constraints. Innovation and collaboration across the value chain will be essential to realizing new opportunities in nutrition, medical, and performance applications [12].

This review aims to assess the potential of whey valorization and to discuss current technological, environmental, and nutritional strategies for its utilization in the context of the circular economy and sustainable development. The topic appears to be important for all concerned sectors, including science, agriculture, and the dairy industry. Over the past decades, the global production of processed whey has steadily increased. This creates the need for new directions in its management. Although considerable work has already been done in this area, including valorization strategies [11,13], as well as whey processing [14,15] strategies, the problem remains urgent, and current solutions are insufficient. The review presents the latest data regarding possibilities of whey by-product utilization, which align with the global sustainable development goals [16]. This review, unlike others published recently, takes into account the latest data from the past decade and broadly discusses the whey potential issue, including extraction, management methods, and use, concerning environmental protection strategies. A novelty is the SWOT analysis, which summarizes the collected data.

## 2. Materials and Methods

All data presented in this narrative review were summarized from the references, including scientific journals and book chapters. The references were searched in the databases PubMed, Web of Science, and Scopus, using the keywords: “whey” AND, “technological process” (OR “processing technology”), AND “utilization” (OR “valorization”). To search for the maximum number of relative references, the keyword was set as “whey and technological process, and/or utilization”, and the search scope was limited to the years 2015 to October 2025. During the period from 1 October 2025 to 5 October 2025, we identified 978 items in these databases, but after analysis, we excluded those (n = 798) that did not meet our inclusion criteria: time period, English language, and that were duplicated in both databases. Ultimately, we selected 164 items for review (Figure 1).

## 3. Results

### 3.1. The Scale of Unused Whey Worldwide and the Associated Environmental Pollution

The increase in milk production is the primary driver of the growing volume of generated whey. Global milk production has risen by 59% since 1988, reaching 944 million tonnes in 2023. This corresponds to the generation of approximately 200 million tonnes of whey annually as an industrial by-product [17]. Depending on the type of cheese produced—such as hard or semi-hard—roughly 1 kg of cheese is made from 10 L of milk, while the remaining 9 L becomes whey, which can reach several million liters per day in large factories [18]. For acid-coagulated cheeses, the ratio is approximately 10 L of whey per 1 kg of cheese. In comparison, soft cheese production, partial moisture, and protein retention may result in a higher percentage of cheese (less whey), and in hard cheese production, the opposite is true. Whey yield in soft cheese production is influenced by the casein concentration in the milk and the whey protein-to-casein ratio, although the latter does not significantly affect coagulation or drainage [18,19,20,21,22,23]. The estimated global whey production and utilization are presented in Table 1.

At an estimated yearly increase of 1–2%, the global volume is expected to rise to about 203–241 million metric tonnes by 2030 [33]. Only 50–60% of this whey is valorized, while 40–50% remains untreated or is used for low-value applications (applications that do not generate high added value and often only serve to dispose of the product, e.g., direct use as animal feed, spraying on fields as fertilizer, or discharging into sewers/treatment plants), posing environmental risks [13,34,35]. As estimated, only 25% of whey is used for direct human consumption, e.g., whey protein concentrates in food products, functional drinks, fat–protein additives, dietary products [29,35]. As such, efforts have been underway to reduce this form of food waste and to valorize this by-product. Valorisation refers to whey products and fractions that have been processed to obtain valuable products—e.g., whey powders (WPC, WPI), protein concentrates, protein isolate, lactose products, industrial preparations (enzymes, cultures), and raw materials for the food and pharmaceutical industries. Valorisation encompasses both food applications and certain industrial products with higher economic value [18].

The European Union and the United States are the largest producers of whey (accounting for ~70%), followed by China (9%), Canada (4%), Australia (3%), and the rest of the world (∼13%) [26,36,37]. In 2023, Germany led the EU in whey production, accounting for 14.1 million tonnes, or approximately 27% of the total. The Netherlands and Poland were the second and third-largest producers, with about 8.8 million and 5.2 million tonnes, respectively [26]. However, the existing literature data on global whey production are limited, fragmented, and often methodologically inconsistent. Reliable and comparable data are largely unavailable, making it difficult to quantify the total volume of whey generated worldwide and to determine the share that is effectively recovered or utilized. Estimation approaches vary considerably, with no standardized tools or conversion frameworks linking milk and cheese production to whey outputs. In this context, the current review addressed gaps concerning methodological limitations of whey production data extraction. The use of methods such as data triangulation or duplicate exclusion would allow for the collection and presentation of the latest and most reliable information on the scale of whey production.

Due to the growing consumer demand for cheese, yogurt, and other dairy products in recent years, the amount of waste from whey production has consequently increased. Dairy production is also highlighted as one of the most energy-intensive sectors with respect to its global warming potential. The high biological oxygen demand (BOD)—which measures the amount of dissolved oxygen that microorganisms consume to break down organic matter in a water sample over a specific period of time—ranged from 30 to 50 g/L in the case of whey, in which sweet whey ranged from 40 to 102 g/L, whereas acid whey ranged between 52 and 62 g/L [2].

The second important indicator of environmental status is chemical oxygen demand (COD), which measures the total amount of oxygen needed to oxidize organic and inorganic matter in a water sample. COD values for whey ranged from 60 to 80 g/L, whereas sweet and acid whey differed (27–60 g/L and 35–51 g/L, respectively). The differences between the whey types are connected with the chemical composition of the samples. Acid whey is lower in lactose in comparison to sweet whey, which may allude to the generally lower COD and BOD values reported in the literature. Regardless that acid whey may have a lower lactose content, its BOD/COD remains high due to increased protein solubility at lower pH [2,38].

Moreover, it should be emphasized that the high BOD and COD of waste whey render it a potent environmental pollutant. Discharging waste whey into the ground, directly into receiving water bodies without pre-treatment, or discharging it into sewage causes environmental pollution, even though cheese whey has been used in agriculture for many years as a fertilizer or animal feed due to its high nutrient content [5].

Land application of whey has been practiced for decades in the USA, Canada, and Australia, as a slow-release nitrogen and phosphorus fertilizer. This practice is generally considered sustainable because whey has been reported to provide an adequate substitute for potassic superphosphate, while some soil properties are improved. However, adverse changes in soil properties, such as salinity, as well as depressed plant growth, can result from high application rates. Sodium accumulation following high-rate whey application is likely to occur at greater soil depths, ultimately affecting future land uses [39]. On the other hand, whey by-product is still identified as a valuable source of nutrients by the paneer industry [33]. Because of that, lactose and milk proteins are the main components causing high BOD and COD levels in whey; it is expected that lactose and protein recovery can help reduce the BOD and COD loads of whey, thereby solving the environmental pollution problem caused by whey disposal [2]. However, this requires further research and will be discussed in detail later in this review.

Recent EU waste regulations, such as Directive 2006/12/EC [40], Directive 91/271/EEC [41], and Directive 1999/31/EC [42], encourage the dairy plants in the EU to valorize whey and dairy wastewater, which usually undergo centralized treatment.

Wastewater in the dairy industry is characterized by:-solids (gross and finely dispersed/suspended);-low and high pH levels;-free edible fat/oil;-emulsified material, e.g., edible fat/oil;-soluble biodegradable organic material, e.g., BOD;-volatile substances, e.g., ammonia and organics;-plant nutrients, e.g., phosphorus and/or nitrogen;-pathogens, e.g., from sanitary water;-dissolved non-biodegradable organics [12].

The rising costs of wastewater treatment have stimulated the growth of the whey valorization market [43]. Dairy companies that recover or use whey significantly reduce the volume of wastewater, which affects BOD (Biochemical Oxygen Demand) and COD (Chemical Oxygen Demand) parameters, the fat content in water, as well as the total nitrogen compounds. For example, from cheese production, when a dairy plant that recovers or uses whey, it significantly reduces BOD by 55%, COD by 74%, fat by 79%, and total nitrogen compounds by 43.4% [12,44].

The bio-economy strategy is one of the major food waste prevention policies in the EU. The Waste Framework Directive [45] prioritizes waste prevention through treatment for reuse (valorisation), recycling, and recovery. Due to its high level of contamination, the valorisation of whey is linked to the achievement of the UN Sustainable Development Goals, such as Goal 6. Clean Water and Sanitation, Goal 12. Responsible consumption and production, and Goal 13. Climate Action [16], as well as the principles of a circular economy (Zero Waste). The use of whey helps reduce water pollution (reduced BOD/COD), and the production of new valuable products [2,46].

It is difficult to obtain detailed information on the scale of whey valorisation due to the large number of dairies, but some authors [47,48] suggest that, for example, valorisation of waste whey can reduce the environmental impact of some cheese production by up to 15%. It should be noted that dairy products have been identified as energy-intensive in terms of their Global Warming Potential (GWP) values [49].

### 3.2. Whey from a By-Product (Disposal) to a Valuable Dairy Product—Changes from the 1950s to the Present

Although significant progress has been made in whey processing over the past few decades, as shown in Figure 2, challenges remain in optimizing these processes for maximum efficiency and sustainability.

1960s–1970s

Cheese was originally manufactured on a small, local scale as a way to preserve milk; consequently, whey was largely considered a waste product. It was discharged directly into water or sewage without pretreatment or stored in tanks. Disposal methods included using cheese whey as fertilizer, which posed a significant environmental risk due to its nutrient content. Contamination of water bodies with whey can lead to the death of aquatic organisms. Moreover, the accumulation of total suspended solids and fat from cheese industry wastewater on the soil damages its structure. Biological treatment, such as anaerobic or aerobic degradation processes, results in a reduction in organic contamination; however, these methods also have many disadvantages, such as the need for specific microorganisms, long hydraulic retention times, or fat flotation [5,6,9].

Whey was primarily used for animal feed, but also for the production of some food products, such as fermented whey beverages. As cheese production grew to a larger scale, this became cumbersome. The factors driving the development of whey protein products varied significantly between the United States, Europe, and New Zealand, where the greatest growth was observed. In the Midwestern United States, environmental concerns related to pollution caused by high volumes of discharge were a major factor driving development in the late 1960s. Estimating a maximum BOD level of 20 ppm for dairy wastewater, regardless of discharge location, was the main reason for the development of the whey processing industry. The drying of whey using roller dryers was introduced to produce so-called “popcorn whey.” This product was characterized by high agglomeration of whey proteins (hence the term “popcorn”), low drying efficiency (whey contains only ~6.5% dry matter), and low commercial value—it was sold primarily as pig feed. This phenomenon was a response to the introduction of a BOD limit for dairy wastewater at the time, which prohibited the discharge of untreated whey. However, this process was inefficient, and the rising cost of natural gas in the early 1970s made this production unprofitable. The development of an economical method for pre-concentrating whey before roller drying initiated the introduction of membrane filtration into the dairy industry [50].

The increase in dairy production has necessitated the introduction of innovative processing methods that enable the effective management of whey. Whey protein products have evolved from the initial production of whey protein concentrate (WPC) and whey protein isolate (WPI) in the 1970s to the diverse range of whey protein products produced today.

1980s–1990s—New whey processing techniques

In the early 1970s, a membrane process known as ultrafiltration (UF) was introduced for whey processing. At the end of the 20th century, other membrane processes, nanofiltration (NF) and microfiltration (MF), were also used for this purpose. This resulted in reduced BOD levels in wastewater and increased interest in value-added whey products within the European industry. More emphasis was placed on the use of ultrafiltration (UF). Denmark was the first country to use this process, producing a lactose-rich product as a fermentation substrate. In New Zealand, the development of beverage companies contributed to the development of whey processing [50,51].

Until the early 1990s, most WPC produced was WPC34/35. It was usually produced in relatively simple spray dryers as a fine powder and used as animal feed or in the food industry. Since the late 1980s, whey protein has attracted considerable attention due to the growing body of research examining not only its nutritional properties but also its bioactive properties [50,51].

At the end of the 20th century, instantized whey protein powder was developed. This product was obtained without added carbohydrates, using the wet agglomeration method, and with improved solubility compared to the prototype developed in 1987. In the 1990s, due to the demand in the sports nutrition market, the demand for high-protein powders increased, which led to their further development [50,51].

21st Century—Next-generation whey products

Currently, the goal of dairy producers is to develop new products and improve traditional dairy items into next-generation products with rich nutritional properties. New technologies enable the separation and extraction of whey protein fractions from whey, through methods such as ion exchange (demineralization) and membrane techniques (reverse osmosis, nanofiltration, ultrafiltration, microfiltration), additionally allowing for the fractionation and concentration of whey [52].

Whey is used as an ingredient to produce complex food products from its waste whey through solids recovery techniques such as nano-, micro-, and ultrafiltration. For example, whey cream with a fat content of 25–30% can be separated from waste whey and used to standardize milk in cheese or butter production or to enrich beverages with protein and essential amino acids.

Cross-flow membrane filtration, including ultrafiltration and nanofiltration, has revolutionized industrial whey processing over the past 50 years. These techniques have enabled the production of WPC and WPI with high protein content and purity. Techniques such as simulated moving bed chromatography and membrane adsorption have improved the separation and fractionation of whey proteins, improving product quality [50,53].

Nowadays, whey processing uses modern, sustainable methods that can lead to the production of new, valuable products in the dairy industry, contributing to the circular economy.

### 3.3. Whey: A Microbiological Overview

The microbiological profile of whey is highly diverse, originating primarily from raw milk, starter cultures, and the processing environment. This is due to the interplay of natural milk microbiota, technological and hygienic factors, and other practices, which determine the safety and functionality of whey-based products.

Lactic acid bacteria (LAB) populations influence not only acidification and flavor development but also the microbiological safety of whey through the production of antimicrobial substances such as organic acids and bacteriocins [54]. Alongside LAB, yeasts represent a significant component of whey microbiota. The dominant genera include species *Kluyveromyces*, *Candida*, and *Saccharomyces,* which are widely used in the biotechnological valorization of whey and in the natural fermentation of whey-based beverages [55,56,57].

The coexistence of LAB and yeasts in whey often results in symbiotic interactions—yeasts release growth factors (vitamins, amino acids) that stimulate LAB growth, while LAB favor low pH values, creating favorable conditions for yeast fermentation. This microbial synergy supports the development of traditional fermented whey beverages and bioactive products [54]. Beneficial LAB species such as *Lactococcus lactis*, *Lactobacillus helveticus*, and *Streptococcus thermophilus* inhibit undesirable microorganisms through acidification, bacteriocin production, and competitive exclusion [58]. A balanced and stable LAB community enhances microbiological quality, maintaining low pH and reducing spoilage or pathogen proliferation. Such whey is a promising substrate for the production of probiotic beverages and as a base medium for other ways of valorization [59].

Whey, a nutrient-rich by-product of cheese manufacture, provides an excellent substrate for microbial growth due to its high content of lactose, peptides, minerals, and vitamins [60]. These components create optimal conditions for the proliferation of LAB and yeasts, which drive two principal types of microbiological transformations—lactic and alcoholic fermentation. Both processes are fundamental for the biotechnological valorization of whey into functional products such as fermented beverages, probiotic cultures, and bioethanol [61,62,63]. The main products of lactic fermentation, such as lactic acid and other organic acids, reduce the redox potential and promote the growth of acid-tolerant beneficial microbiota [31].

Conversely, the ethanolic fermentation of whey, carried out under either aerobic or anaerobic conditions depending on the desired end product, is primarily driven by *Kluyveromyces marxianus*. This yeast possesses β-galactosidase activity, enabling the hydrolysis of lactose into glucose and galactose, which are subsequently fermented to ethanol [64]. Under controlled fermentation conditions, *K. marxianus* also produces volatile esters, aldehydes, and organic acids that contribute to the flavor profile of fermented whey beverages and play a key role in whey valorization [65].

The efficiency of these bioconversion processes depends on the precise control of key fermentation parameters, such as temperature, pH, inoculum level, aeration, agitation, and fermentation duration, which collectively influence microbial growth dynamics, substrate metabolism, and overall metabolite yield [66,67]. Nevertheless, variations in the native microbial population may lead to inconsistent fermentation performance and reduced process predictability. Therefore, in industrial applications, the natural microbiota must be carefully monitored or standardized to ensure reproducible fermentation efficiency and product safety [68].

Another representative group of the health-promoting metabolites occurring in whey is organic acids, with lactic acid being a dominant compound. However, during lactic fermentation, LAB also produce acetic acid and phenyllactic acid (PLA), which results from the metabolism of aromatic amino acids, among other compounds. These organic acids lower the pH, disrupt cell membranes, and chelate metals, all of which contribute to antimicrobial and antifungal effects. Additionally, they enhance the antioxidant capacity of fermented whey products [23,69].

The low pH of acid whey inhibits the growth of many pathogens, but does not guarantee full inactivation. Acid-resistant strains, such as *Escherichia coli* O157:H7 and *Listeria monocytogenes,* can survive in low-pH fermented dairy. While growth is unlikely in acid whey, the survival of contaminants at refrigeration temperatures is possible for weeks [70,71]. Additionally, whey and whey products can be contaminated by aerobic mesophiles, coliforms, enterobacteria, and microorganisms that might come from diseased animals [72].

In summary, lactic fermentation of whey offers strong advantages in microbiological safety because LAB produce organic acids and bacteriocins that lower pH, inhibit spoilage organisms, and support the development of probiotic and functional products. Its main limitations are that low pH does not fully eliminate acid-resistant pathogens, and natural microbiota variability can reduce process consistency. In contrast, alcoholic fermentation—driven mainly by *Kluyveromyces marxianus*—efficiently converts lactose to ethanol and generates desirable flavor compounds, but it is more sensitive to fermentation conditions and offers less intrinsic microbial inhibition than lactic fermentation. Both methods enable valuable biotransformation of whey, yet each has trade-offs between safety, product quality, and process robustness.

### 3.4. Whey Proteins and Other Whey Constituents, and Their Functionality

Biotechnological processes are considered an attractive direction for managing and processing whey. Whey, as a by-product, is a cheap substrate for various methods and is also very valuable due to its composition. The selection of appropriate microorganisms capable of transforming the components contained in whey (mainly lactose) and the proper process conditions allows the production of valuable products, most frequently used in the food and pharmaceutical industries [3,73]. Whey is also used in animal feed, in meat production, in the production of dietary foods and high-protein supplements, as a source of potentially probiotic lactic acid bacteria (*Lactobacillus* sp. and *Bifidobacterium* sp.), in biotechnology, and for enriching food and pharmaceutical products. Whey is also used in the confectionery, bakery, pastry, and dairy industries. These activities help reduce food waste and protect the environment [74,75].

Whey protein is widely used to increase the biological value, improve textural, physical, and other functional properties of foods, enhance sensory characteristics, and formulate low-lactose, high-protein products. Changing consumer preferences are driving food and beverage manufacturers to add functional whey protein to their products, as it can replace high-fat and expensive ingredients like cream cheese, milk, and butter while maintaining the original appearance, flavor, and texture of the products. Whey protein is used in the food industry in baked goods, dairy products, beverages, breakfast cereals, chocolate, and baby food for its antihypertensive and antibacterial properties [74,75]. Whey protein hydrolysate (WPH) has antimicrobial, antioxidant, and antihypertensive properties. It is used in the production of clinical nutrition, sports nutrition, and infant nutrition products. WPH is also used in the production of bakery products, low-fat foods, dietary foods, and protein-fortified beverages. Consumer preferences for incorporating protein into everyday foods, along with the growing demand for high-protein, low-fat products to support weight management, are expected to drive growth in this segment. WPC offers an affordable alternative to caramel mixes, boasting excellent taste and processability. WPC is used in various applications, including the production of yogurts, dairy drinks, and desserts. Furthermore, these concentrates are used to enrich infant formulas and food products with protein. When heated and dissolved in water, WPC exhibits gelling properties, which are useful in applications in the meat and food industries [74,75]. Table 2 presents whey protein preparation and its application.

### 3.5. Novel Whey Processing Methods

Sustainable whey processing techniques, which provide important value-added products for the dairy industry, include:-Thermosonication—used to produce whey beverages, including fermented whey beverages. This method offers better microbiological quality and enhanced antioxidant, antihypertensive, and antidiabetic activities; reduces the degradation of ascorbic acid, improves rheological properties, and provides a more favorable volatile compound profile when compared to beverages treated with short-term thermal heating.-Ohmic heating and spray drying—used in the production of whey protein, such as WPH, WPI, WPC.-Cross-flow microfiltration—used for concentrate whey as native powder, deproteinized powder, or delactozed powder.-Electroactivation—whey lactose is used to produce galactooligosaccharides (GOS), lactitol, lactulose, and lactosucrose.-Reverse osmosis—whey water for water recovery.-Membrane technologies (microfiltration, ultrafiltration, nanofiltration, emerging membrane processes, integration of membrane technologies)—used for clarification or fractioning, as well as to increase the concentration of specific components (concentrates whey proteins and decreases non-protein components), or their separation. Moreover, this process leads to a highly efficient separation of phenolic compounds while ensuring bacterial reduction. This results in high-value whey protein concentrates (WPCs) and isolates (WPIs), which are rich in essential amino acids and ideal for sports nutrition and dietary supplements.-Enzymatic hydrolysis—improves the functional properties of whey proteins through controlled cleavage of peptide bonds, which reduces molecular weight and increases the exposure of polar groups; and improves solubility, emulsification ability, and foam stability. Hydrolysates are ingredients in functional foods, beverages, and products with regenerative properties.-Unconventional technologies, including intensive ultrasound, cold plasma, microwave, supercritical carbon dioxide, and ohmic heating—a non-thermal and gentle method that helps preserve the physicochemical, nutritional, and sensory properties of whey [14,50,51,52,88,112,114,115,116].

These innovations and modern technologies enable the production of high-purity whey protein concentrates, isolates, hydrolysates, and fractions with a wide range of applications and other specialized products that address the increasing market demand for whey-based components. This also includes the production of whey powders, edible films, and biofuels, which further contribute to a more sustainable dairy industry by maximizing resource utilization [3].

It is worth mentioning that the use of various techniques of whey valorisation has its advantages and disadvantages. These techniques are characterized by different efficiency, operating costs, and environmental impact, which requires informed selection depending on the desired end product.

For example, membrane separation techniques such as microfiltration, ultrafiltration, and nanofiltration are more economical and environmentally friendly than traditional methods. Because they operate at low temperatures, they consume less energy than thermal methods and provide high separation selectivity [117]. Currently, they constitute the most advanced group of methods in terms of scalability, separation efficiency, and relatively low energy consumption. However, their main limitation remains membrane clogging and the need for regular regeneration. These techniques ensure the recovery rate of 50–55% of whey protein (WP) with better functional quality (recovery of an individual fraction of protein with a purity of 90%), and are relatively simple, cost-effective, and consume less energy compared to other methods [116].

These methods cannot be used to valorize acid whey due to differences in composition (higher acid content and lower pH) that affect the behavior of lactose, which tends to crystallize, making filtration difficult. The advantage of this method is its relatively low cost compared to chromatographic techniques and mild heat treatment, which preserves bioactive compounds [118,119,120]. To obtain demineralized whey, it is concentrated by evaporation or reverse osmosis (RO), and then is processed using ion exchange columns. These processes are used on an industrial scale but require high capital and operational costs, both for infrastructure and for maintaining column regeneration systems, which affects the total production cost [121].

Compared to these methods, new-generation technologies—such as thermosonication (TS), cold plasma, intense ultrasound, and supercritical carbon dioxide (SC-CO_2_) —stand out for their ability to preserve bioactive compounds and improve the sensory quality of the product. However, their implementation is limited by higher energy consumption and equipment costs and a lack of full process standardization. For example, the disadvantage of the TS method is scalability, as it requires expensive equipment and may not be suitable for large-scale production [122,123].

Enzymatic hydrolysis, on the other hand, is a method with lower energy requirements and high precision control of protein functionality, but is associated with costs related to the purchase of enzymes and requires longer processing time [102,124].

Technologies such as reverse osmosis (RO) and spray drying play a key role in recovering water, ingredient concentration, and the production of stable whey powders that can be used in further stages of food production. Reverse osmosis is one of the most sustainable methods for reducing water waste, while spray drying is a technology characterized by high production efficiency, although requiring relatively high energy consumption. Ohmic heating is a highly effective technology for processing WPC, WPI, and WPH whey proteins thanks to the rapid and uniform heating of the product. However, it requires higher investment costs, which may limit its use in smaller-scale production plants. Electroactivation and other lactose conversion processes enable the production of high-value products, such as galactooligosaccharides, lactitol, and lactosamine. However, these technologies remain niche, primarily due to the limited number of industrial implementations and the need for strict control of process parameters. Membrane technologies and integrated processing systems (e.g., ultrafiltration with reverse osmosis and spray drying) seem to be the most effective from a sustainability perspective and the most cost-effective approach for large-scale industrial applications. They enable water recovery, ingredient concentration, and waste reduction, making them the most sustainable technologies throughout the entire production cycle [125].

Unconventional technologies offer particular advantages in terms of product quality, although they require further research and optimization on an industrial scale. The use of ultrasound increases the permeation rate in the ultrafiltration process by minimizing the concentration of polarization and reducing membrane fouling [2,122,123,124,126,127,128,129,130,131].

This comparison highlights the typical trade-offs in whey processing—between operating costs, energy consumption, product quality, and scalability. Therefore, the choice of whey processing technology should be tailored both to the raw material parameters (e.g., sweet vs. acid whey) and the expected technological effect.

### 3.6. The Utilization of Whey for the Production of High-Quality Food Products

Whey is utilized in different ways across various countries, reflecting its nutritional value and versatility. It is used in animal nutrition, meat production, dairy production, dietary foods, and high-protein supplements, for the production of whey beverages, as a nutraceutical, as a source of potentially probiotic lactic acid bacteria, in biotechnology, for the production of whey protein preparations, as a standalone product, and for the fortification of food and pharmaceutical products [7]. The selected utilization methods are discussed below.

#### 3.6.1. Whey Cheese Production

Whey is mainly used to produce whey cheeses, which are popular in many regions due to their unique taste and texture (Table 3). Some of them, due to their proven connection to the region’s history and culture, have been designated as Protected Designations of Origin (PDO) [132].

Cheeses are made by heating whey to denature its proteins, resulting in products with high moisture content and pH levels that are susceptible to microbial growth and require appropriate preservation techniques. Each country utilizes whey in its own unique way, reflecting local traditions and culinary preferences [86,133,134,135].

Cheese whey can be processed into secondary products such as ricotta. This process also produces secondary biowaste from cheese whey, the disposal of which is prohibited using traditional methods in the EU. The use of waste whey as animal feed is currently not recommended and not practiced due to its high lactose content and the potential acidification of waste whey [136].

#### 3.6.2. The Use of Whey in Meat Preparation

Acid whey (AW) [137] is a source of whey proteins and lactic acid bacteria (LAB), with antimicrobial and antioxidant properties. Using whey to prepare meat may help to reduce the amount of preservatives added to food and extend the shelf life of various types of meat. Probiotic bacteria can be used as starter cultures in meat marinades. Marinating meat in whey inhibits oxidation processes and improves its color stability after heat treatment, physicochemical properties (tenderness, juiciness), and the taste of meat products, as indicated by numerous studies [138,139,140,141,142,143,144,145]. The quality of meat marinated in acidic whey depends on the duration of the process. The optimal time is 24–72 h; extending or shortening this time negatively affects the palatability of various types of meat, or does not affect its quality [137,141,145,146]. According to the authors, the effectiveness of acid whey in tenderizing meat is greater in muscles with a high content of connective tissue. Proteolysis by meat proteases (calpains and cathepsins) and endopeptidases derived from acid whey may affect its water-binding capacity. The action of lactic acid bacteria in meat results in better colonization and activation of the enzymatic mechanisms of lactic acid bacteria, including proteases [147]. Marinated meat contained more peptides and a small amount of biogenic amines due to the presence of *Lactobacillus* bacteria in whey, which have a proven ability to produce amine oxidase enzymes and degrade biogenic amines [148].

#### 3.6.3. The Valorization of Whey in the Production of Functional Beverages

Whey is recognized as a versatile and sustainable raw material for beverage production due to its unique nutritional composition, functional properties, and technological potential. As a by-product of cheese manufacture, whey contains lactose, whey proteins, minerals, vitamins, bioactive peptides, branched-chain amino acids (BCAA), low-fat content, and a relatively neutral flavor. These make whey an attractive ingredient in the beverage sector, and an ideal matrix for formulating various types of beverages, including fermented, non-fermented, functional, and protein-enriched drinks. Whey and whey components have been used commercially to produce whey-based beverages, regular and supplemented fruit juices, milk or milk permeate, as well as nutraceutical compounds and/or probiotics/prebiotics, for example, naturally carbonated whey-based probiotic beverages [54,85,112,149,150,151].

Whey kefirs are produced from kefir grains containing LAB and yeasts (*Lactobacillus kefiri, Kluyveromyces marxianus*). They are characterized by a slightly acidic flavor, mild effervescence, and probiotic activity [152].

Similarly, Saha et al. [153] demonstrated the effective use of chhana whey in fermented beverages prepared with *Streptococcus thermophilus* (NCDC-74) and yogurt (TC-470) cultures. Furthermore, Rizzolo & Cortellino [154] developed a whey-based beverage combining ricotta cheese whey with fruit juice, highlighting whey’s potential as a functional ingredient. On the other hand, whey–fruit beverages, in which whey is blended with fruit juices (e.g., orange, apple, or berry juices), are creating refreshing drinks with improved flavor, nutritional value, and antioxidant potential [155,156]. Meanwhile, whey–cereal beverages—in which whey is combined with barley, oats, or rice extracts to provide a natural source of dietary fiber, β-glucans, and prebiotic compounds—are valued for their probiotic potential and ability to support gut health [157,158,159].

Fermented whey drinks are considered functional beverages, as they often contain beneficial bacteria, including probiotic strains that can improve intestinal microbiota balance, enhance immune function, and contribute to overall well-being [159]. The use of innovative, beneficial bacterial strains in the production of whey beverages is both practically and commercially promising, as it enables the closure of the raw material cycle, creates valuable functional products, and crosses into the “health-promoting” beverage segment. However, the key is a combination of factors, which include strain selection (good matrix compatibility), cell protection technology (microencapsulation/stabilizing formats), and validation of health benefits. Only a limited number of health claim applications have been approved under the current food law. Adapting the law to the activities of food producers, along with creating a reliable database of the probiotic properties of specific microorganism strains and the technological requirements that enable these effects to be maintained throughout the product’s shelf life, poses a significant challenge [160]. On the other hand, basic guidelines on probiotics have been established by the Food and Agriculture Organization (FAO) and the *Codex Alimentarius*. In the United States, probiotics are primarily regulated by the Dietary Supplement Health and Education Act (DSHEA) and the Food and Drug Administration (FDA). In UE the EFSA/Regulation (EC) 1924/2006 [161] framework established rules for nutrition and health claims on food products in the EU, which are based on scientific evidence and must not be misleading. Taking into account existing regulations and FAO/WHO probiotic guidelines, it can be assessed that there are no clear regulations regarding the declaration of the probiotic effects of a food product containing beneficial microorganisms [161,162]. Therefore, the use of probiotics as starter cultures to produce whey beverages that provide health benefits is not widely adopted in the industry and is currently limited to the scientific sphere.

Non-fermented whey beverages are typically formulated by flavoring or fortifying clarified whey with natural ingredients such as fruit juices, plant extracts, or dietary fibers. These products maintain the nutritional value of whey [163]. Functional whey beverages enriched with amino acids, vitamins, and plant-derived ingredients provide nutritionally balanced solutions [164,165]. A recent innovation in the formulation of whey-based beverages involves the incorporation of superfood ingredients such as chia seeds, matcha, and turmeric, which enhance both the nutritional profile and consumer appeal of these products. To obtain a smooth, creamy texture without relying on artificial stabilizers, natural hydrocolloids such as guar gum and xanthan gum are commonly used [109,166,167].

In summary, recent advances in product formulation have enabled the development of a wide range of whey-based beverages, including protein shakes, smoothies, and fermented drinks. Combining whey with fruit juices, plant extracts, or natural flavors creates new market opportunities and attracts consumers seeking alternatives to traditional dairy beverages. Whey is increasingly being used in the development of functional foods and beverages, including sports drinks and fermented beverages.

#### 3.6.4. Other Usage of Whey

The reason for the continuous development of whey processing is the pursuit of sustainability. Cheese whey and whey permeate are increasingly utilized as substrates for energy production due to their high organic content. The main applications include bioethanol, biogas, and microbial fuel cells production [168,169,170]. Utilizing whey to produce biofuels, biohydrogen, bioplastics, bioethanol, single-purpose proteins, and other value-added products addresses environmental concerns associated with whey disposal and promotes sustainable biorefining practices [99].

Whey cream has gained popularity due to its various applications in the food industry, including butter processing, as well as in cosmetics, owing to its high protein and nutrient content, making it a valuable ingredient [171,172]. Whey protein may also be an alternative to iron fortification in foods and supplements. Research indicates that whey protein can effectively chelate iron, improving its absorption and stability in various formulations. Peptides released during alcalase-mediated hydrolysis of whey proteins under optimal conditions exhibit potent iron-chelating properties. The chelation efficiency, under optimal conditions, is approximately 77.38% effective. Peptide-iron complexes derived from whey are potentially new functional ingredients that can be used as a carrier of bioavailable iron [104,173].

##### Edible Whey Films

Edible whey films are an innovative food packaging solution, providing a sustainable alternative to traditional synthetic materials. They can be made from whey protein isolates (WPI) and whey protein concentrates (WPC). New WPI-based nanocomposites can be incorporated into multilayer film packaging. They are biodegradable, flexible (mechanical properties), and can be enriched with various additives to improve their functionality. The development and use of whey-based films stems from the need for environmentally friendly packaging solutions that maintain food quality and safety. These films are characterized by high water vapor permeability and excellent oxygen permeability (barrier properties), making them suitable for food packaging applications [28,107,109].

Furthermore, bioactive and antimicrobial whey proteins, such as lactoferrin, can be incorporated into edible films, creating functional packaging that can extend the shelf life of products. Lactoferrin is effective against Gram-positive and Gram-negative bacteria, yeasts, and viruses. Therefore, it can potentially be used in a wide array of food products [81]. The use of single-cell protein (SCP) derived from whey to produce edible films is a novel approach that enhances the circular economy by utilizing agricultural byproducts [107]. Various additives are used to improve properties such as water vapor permeability and antioxidant activity. Egg proteins enhance the thickness, dissolution time, and protein content of whey films, thereby improving the physical and chemical properties of the whey films. Chia seeds improve mechanical properties. Ginger essential oil enhances the antimicrobial properties of whey films, effectively inhibiting the growth of pathogens, such as *Escherichia coli* and *Staphylococcus aureus* [96,174,175,176]. Antimicrobial coatings for cheese are made from whey protein–polysaccharide. Antifungal plastic films carrying Lactic acid bacteria fermented whey for the preservation of cheese slices [28,177,178,179].

##### Alternatives to Petroleum-Based Plastics

Other examples include polylactic acid (PLA) and polyhydroxyalkanoates (PHA)—bio-based polymers that serve as environmentally friendly alternatives to petroleum-based plastics. PLA and PHA are produced through the anaerobic digestion of whey [98,111]. These types of biopolymers are used in packaging materials, disposable products, agricultural films, and even medical applications.

##### Whey-Derived Alcoholic Beverages

The lactose content of whey enables its fermentation into ethanol and subsequent distillation into whey-derived alcoholic beverages [180]. Researchers achieved a 2.56% *v*/*v* ethanol concentration by optimizing lactose concentration and fermentation time using the yeast *Kluyveromyces marxianus*. This process, conducted under low aerobic conditions (0.4 vvm), resulted in a quantified ethanol yield of 92.92% using the specified strain [181]. Moreover, acid whey is used as a substrate for the production of Lactiobionic acid by *Pseudomonas taetrolens.*

##### Biohydrogen and Biodiesel

Cheese whey fermentation utilizes microorganisms to convert organic substrates into hydrogen. It is an indirect technology that employs several types of bacteria (Clostridium and Enterobacter) to produce biohydrogen and volatile fatty acids (VFA) under anaerobic conditions, taking advantage of the high carbohydrate content in waste whey. The fermentation process typically occurs in two stages, during which lactate is first produced and then metabolized to hydrogen. *Lactobacillus* is the dominant genus in whey fermentation, primarily converting carbohydrates into lactate, which is then utilized by hydrogen-producing bacteria, such as *Clostridium.* However, this process is difficult to control and sensitive to substrate composition. This solution is promising as it enables the production of renewable H_2_, with significant potential for large-scale implementation compared to alternative biological processes due to the use of renewable substrates, process simplicity, and independence from light [182,183,184,185]. Studies have shown that VFAs can be used as a low-cost alternative carbon source that could be converted into lipids and then into biodiesel [51,186,187].

The conversion of whey into bioethanol, biogas, and biohydrogen is carried out under different conditions and leads to varying degrees of environmental pollution reduction, as indicated by COD values. The yields and operating conditions presented in Table 4 show that whey is indeed a viable substrate for bioethanol, biogas, and biohydrogen production. TEA and LCA data support the sustainability and economic potential of whey valorization. If managed well (especially with “gate-fee” credit), the process can be cost-competitive and reduce environmental impact. However, achieving high yields often requires careful control of fermentation parameters (temperature, pH, COD concentration), and the scale-up economics strongly depend on local conditions (whey disposal costs, energy prices, capital investment).

### 3.7. Technological Problems with the Valorization of Whey

Problems related to the valorisation of whey include:raw material variability;process challenges depending on the type of whey (acid vs. sweet whey);microbiological safety/quality;economic aspects and scalability;environmental/regulatory issues.

Milk whey poses a significant threat to the environment if disposed of untreated. These types of issues may be solved with innovative cavitation technologies and devices enabling the extraction of milk raw material from industrial wastewater and its processing for feed purposes. The most common issues related to processing milk whey are technological problems associated with raw material variability, isolating proteins from liquid, its quality, safety, and environmental concerns [192].

The main obstacles in retaining whey protein are the high variability in the composition and properties of whey-derived products. Based on the manufacturing process, whey can be separated into two categories: acidic cottage cheese whey (with a pH range of 4.35–4.41) and sweet rennet cheese whey (with a pH range of 6–7) [23]. In sweet whey, glycomacropeptide can be removed by whey separation before cheesemaking using microfiltration. On the contrary, valorization through membrane filtration of acid whey could be challenging. Acid whey can be treated by nanofiltration, diafiltration, reverse osmosis, and electrodialysis for demineralisation or lactose recovery. However, crystallization of lactose is mainly a drying and storage issue [2,53]. Processing highly acidic whey poses a challenge due to the significant complexity of acidic whey processing. Its deoxidation with alkaline reagents leads to the disruption of quality in processed products, such as changes in taste and possible chemical contamination [192].

In various forms of whey, such as concentrated, evaporated, or dried whey, there are additional concerns regarding the presence of spores, particularly those of *Bacillus cereus*, as well as heat-stable and dry-stable toxins and enzymes. Notably, spore-formers, particularly the *Bacillus cereus* group, can endure pasteurization, persist in whey processing lines, and become concentrated in evaporated or condensed whey and powders [193].

In acid whey, the main hazards involve the survival (rather than growth) of acid-tolerant pathogens, as well as issues related to post-process spoilage and biofilm formation [193]. The wide range of possibilities for using whey as a matrix for producing fermented and non-fermented beverages is associated with several technological challenges. Choosing the right technology is crucial to minimize any negative effects on the rheology of whey, including losses in bioactive compounds (such as peptides, organic acids, or other microbial metabolites), and maintain the sensory quality of final whey products.

Despite its significant potential, the production of whey beverages can encounter some technological barriers:-Microbiological stability—Whey provides an environment conducive to microbial growth, so when designing these types of beverages, it is essential to develop effective preservation methods, such as pasteurization, microfiltration, and the use of natural growth inhibitors.-The sensory quality of the developed whey beverages: The natural flavor and aroma of whey are not always accepted by consumers; therefore, the use of flavorings or fermentation processes to enhance organoleptic qualities is crucial. Fruit additives are particularly effective in this regard.-The high lactose content limits the consumer group, excluding those with intolerance. Enzymatic hydrolysis of the disaccharide lactose should be considered when planning and selecting production technologies.-Standardization of the final product—The wide variation in whey composition depending on the type of raw material, the seasonality of the milk, and the cheesemaking process makes it difficult to achieve a uniform final product. Technological costs associated with implementing new whey processing technologies (e.g., ultrafiltration, nanofiltration) are also significant, as they necessitate investments in modern production lines, which may be a barrier for smaller enterprises.

Paneer whey was added to the production of multigrain bread, leading to enhanced crust browning linked to the Maillard reaction, along with a higher total solid content. Milk and acid whey-containing samples presented higher crumb firmness, gumminess, chewiness, and resilience compared to the control, as observed by Paul et al. [194]. It is possible to enhance the nutritional value of bakery products by incorporating high-protein ingredients, such as whey protein. However, if the whey protein content exceeds 5%, a bitter taste is observed due to the presence of small peptides, ferulic acid, and tannins [195]. On the other hand, the beverage industry faces challenges in utilizing whey protein due to the development of off-flavors and bitter peptides [97]. Fancello et al. [11] also found that fermented beverages with second cheese whey were associated with rancid odor caused by free fatty acids, esters, and ketones from whey. Liu et al. [196] used LC-TOF-MS/MS analysis to identify and characterize tri-to nonapeptides as bitter peptides and suggested that ultrafiltration could be used to remove these bitter peptides. Although positive results for the effective removal of bitter peptides by ultrafiltration were obtained, they were very limited. An extensive study should be conducted to develop more economical and commercial techniques for removing bitter peptides from the product without altering its nutritional and functional properties [97].

Whey can be a source of heavy metals, which is another technological problem. According to de Aquino et al. [197], there was a significant variation in the levels of mercury (Hg) detected. The highest concentration observed was 9.41 ± 0.295 ng g^−1^, while the lowest level recorded was 0.548 ± 0.029 ng g^−1^. The concentrations were determined to be within the acceptable range for food products. However, there are currently no worldwide regulations regarding the presence of mercury contamination in dairy products.

## 4. Future Perspectives and Conclusions

Currently, in industrialized countries around the world, 70–90% of whey is processed for various purposes. However, the complete and rational use of whey has not yet been achieved worldwide. In many industries, residual whey, which constitutes the majority of the by-products, is discharged into the sewer system without adequate treatment, leading to substantial environmental degradation. This practice is primarily the result of a lack of advanced processing technologies, inadequate economic incentives, and the high cost of building processing plants [163].

On the other hand, the global whey protein market was estimated at USD 5.33 billion in 2021 and is expected to reach USD 14.32 billion by 2030, growing at a compound annual growth rate (CAGR) of 10.48% between 2020 and 2030. The whey protein concentrate (WPC) segment accounted for the largest share of global revenues in 2021, exceeding 40.00%, while the whey-based sports nutrition segment reached 21.50%. Due to its versatile use in the food industry, global whey protein sales are expected to grow, as consumers continue to seek functional food products, including those addressing issues such as lack of appetite, protein-calorie malnutrition, and disease-related muscle loss. Demand will also be influenced by the development of fortified food products, edible films, cosmetics, and skin care sectors [74].

We used a SWOT analysis to identify the strengths, weaknesses, opportunities, and threats related to whey valorisation (Table 5).

Failure to valorize whey by-products leads to significant economic, nutritional, and environmental losses. New directions for whey processing are still being sought, but some of the whey remains unused. Potential processing directions include using whey as a source of bioactive peptides, probiotics, organic acids, aromatic compounds, and enzymes, with potential applications for the production of value-added products through microbial fermentations. An increasing awareness of health and fitness has fueled a growing demand for protein-rich foods and other functional foods, further driving the development of this food segment.

A promising direction is the integration of whey processing into a circular economy, where waste is transformed into a valuable resource. These approaches not only reduce environmental burdens but also contribute to the sustainable development of the agro-industrial complex. Deep whey processing presents new opportunities for developing eco-friendly technologies across various industries, minimizing environmental risks, and creating new markets for highly profitable products. The rational use of whey is therefore a key factor in the innovative development of the dairy industry and the economy as a whole. Whey is now a key resource with enormous potential for innovation, sustainability, and economic growth through advanced processing and integration into the circular economy [51].

Progress in whey processing is promising, but challenges remain in optimizing its properties to enable its use on an industrial scale. The incorporation of whey into various sectors requires ongoing research and development to improve its functional properties and economic viability. Future research should focus on developing cost-effective strategies for membrane technology, standardization of whey composition, as well as the use of whey as a carrier of bioactive compounds for the development of functional foods.

## Figures and Tables

**Figure 1 foods-14-04258-f001:**
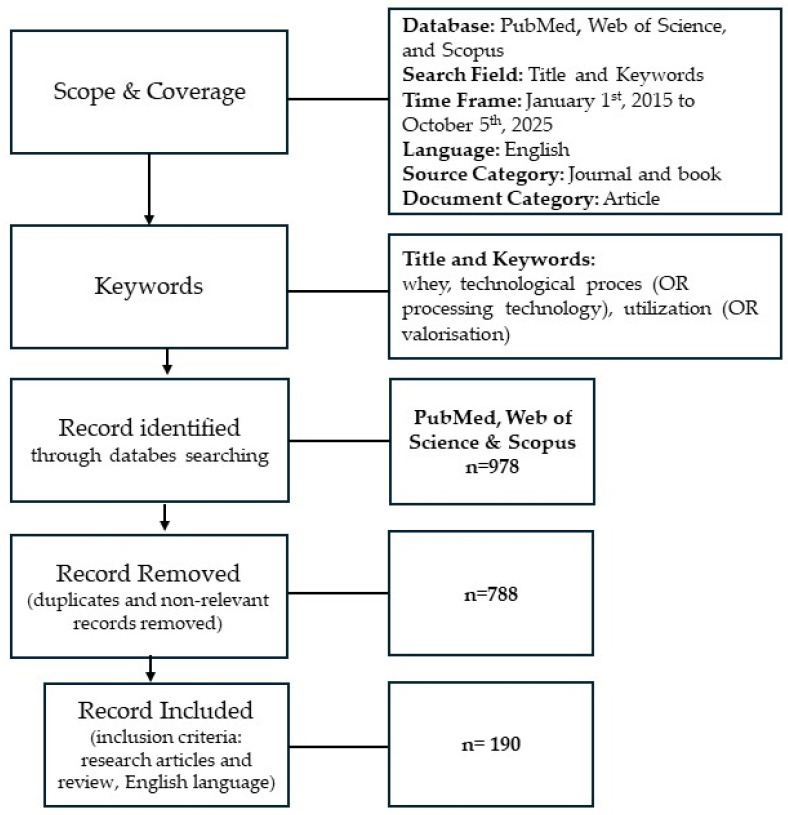
Flow diagram of the literature search and selection process.

**Figure 2 foods-14-04258-f002:**
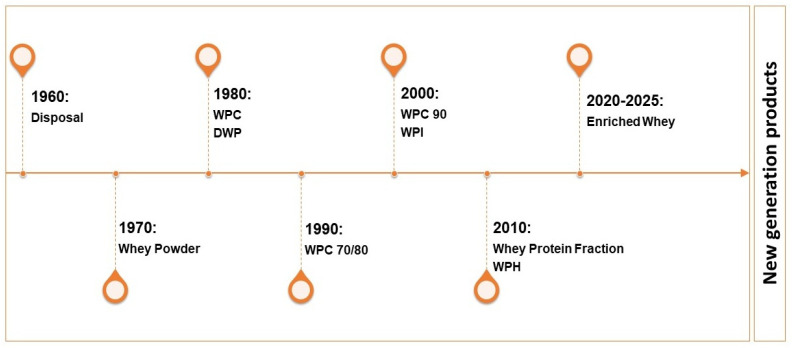
Evolution of whey processing (1960–2025). Explanatory Note: WPC—Whey Protein Concentrate, DWP—Demineralized Whey Powder, WPI—Whey Protein Isolate, WPH—Whey Protein Hydrolysate.

**Table 1 foods-14-04258-t001:** Global Raw Liquid Whey Production and Utilization by Region.

Region	Annual Whey Production (Metric Tons)	Notes	Reference
European Union	52.2 million	Used for human consumption	[24,25]
in which;			
Germany	14.1 million	By-product in the manufacture of cheese	
Netherlands	8.8 million		[26]
Poland	5.2 million		
United States	n/d (fragmented estimation > 58.2 million *)	Sweet whey estimated from USDA data	[27,28]
Argentina	4.7 million (4700 million liters)	Processed into demineralized powder and whey protein concentrates	[29]
India	3.3–5 million	Estimated from paneer production	[25,30]
Jalisco ** (in Mexico)	0.323 million (322,942 m^3^)	High organic load, environmental impact	[31]
Turkey	1.1 million	As a possible energy source	[32]
Global	180–200 million	Significant environmental impact due to high biological oxygen demand	[13,19,21,22]

* Liquid whey originating solely from cheese manufacture (excluding cottage cheese) is estimated on USDA data [27] and a conversion of 9 L of whey per 1 kg of cheese [28]. ** Jalisco is a major dairy-producing state of Mexico. Own study.

**Table 2 foods-14-04258-t002:** Whey proteins and other constituents, functional properties, and applications.

Type of Whey Preparation	Processing Methods	Functional Properties	Chemical Properties	Technological Application	Source
Whey protein concentrate (WPC)	Ultrafiltration, Diafiltration, Spray Drying	Good solubility, high emulsification and foaming capacity, moderate heat stability, mild milky flavor. WPC hydrolysates possess antioxidant and ACE-inhibiting effects. Peptides released from WPC after hydrolysis have bioactive properties.	Protein content 35–80%; moderate amount of lactose and fat; calcium and phosphates content.	Sports nutrition, sauces, milk drinks, baby food, protein bars, functional drinks, confectionery, desserts, meat products, bread	[28,63,75,76,77,78,79,80,81,82,83]
Whey protein isolate (WPI)	Ultrafiltration; Microfiltration; Ion exchange; Spray drying	Very good solubility; excellent emulsifying properties; high thermal stability; neutral taste. In soluble WPI dispersions, the addition of bioactive extracts (e.g., polyphenols) can modify the physicochemical properties (viscosity, interactions).	Protein content ≥90%; very low lactose (<1%) and fat content; high purity of β-lactoglobulin and α-lactalbumin fractions	Protein supplements, dietary products (lactose-free), clinical nutrition, functional drinks	[52,77,79,84,85,86,87,88]
Whey protein hydrolysate (WPH)	Enzymatic hydrolysis (e.g., papain, trypsin, alcalase); Heat inactivation; Spray drying	They exhibit high solubility, rapid absorption, and bioactive properties, including antioxidant, ACE inhibitor, and immunomodulatory effects. They possess reduced foaming capacity; very good heat stability; a slightly bitter aftertaste; and high biological value (DIAAS > 1.0). Peptide fractions (e.g., <1 kDa) demonstrate stability and bioactivity after simulated digestion.	High protein content (75–85%) and bioactive peptides; very low lactose content	Formulas for infants and the elderly, medical nutrition, regenerative supplements, and also in muscle systems (e.g., improving protein stability in meat).	[89,90,91,92,93,94,95]
Demineralized whey powder (DWP)	Nanofiltration, Electrodialysis, Spray drying	Good solubility; improves the taste and structure of products; has stabilizing properties	Low ash content (<5%), moderate lactose level, protein content approx. 12–15%	Production of infant food, dairy desserts, protein bars, instant soups, and sauces.	[52,88,96]
Single-cell whey protein (SCP)	Microbiological fermentation (*Bacillus*, *Kluyveromyces*), drying, grinding	Highly digestible; nutritional function; possible use as a source of amino acids	Protein content (60–70%); trace element content (Fe, Zn, Mg)	High-protein products, feed, and biotechnology applications	[63,97]
Whey permeate powder	A by-product of whey protein ultrafiltration	Nutritional functions, source of protein, and oligosaccharides	Protein content 3–8%, content of lactose, minerals, and oligosaccharides	A cheap raw material for fermentation (lactic acid, bioethanol) and the production of bioplastics, feed, and fertilizers; a source of substrates for biotechnology.	[28,83,98,99,100]
Prebiotic products from lactose (GOS, lactulose)	Enzymatic conversion of lactose (β-galactosidase, isomerization) to GOS and lactulose	Prebiotic properties: stimulate the growth of intestinal bacteria (*Bifidobacterium*, *Lactobacilus*)	GOS content up to 60%; thermally stable; supports the bioavailability of calcium and magnesium	Used as a functional ingredient for a wide range of products: infant milk powders, dairy products, fruit-based drinks, bakery products, and prebiotic supplements. Lactulose is used as a sweetener for diabetic patients, replacing the sugar in confectionery and yogurts.	[52,99,101,102,103,104,105,106]
Whey proteins	They are produced enzymatically or by the isomerization of lactose.	They have prebiotic properties and support intestinal microbiota. WPI/WPC can be used as a film matrix; modifications improve mechanical properties and antimicrobial activity.	They contain: β-lactoglobulin, α-lactalbumin, immunoglobulins, and lactoferrin	Used in the production of meat and meat products, reduced-fat products, yogurts and ice creams, cheeses, bakery products, confectionery and pastry products, infant formulas, and whey drinks. Used as edible films, protective coatings, and biodegradable packaging, is used in functional foods, supplements, protein drinks, and clinical products.	[8,51,52,67,102,107,108,109,110,111,112,113]

**Table 3 foods-14-04258-t003:** Examples of whey cheeses, traditionally produced from milk whey, include those recognized under PDO/PGI status in the European Union.

Country	Types of Cheese	Country	Types of Cheese
Italy	Ricotta salata, Ricottone, Ricotta fresca Ricotta di Bufala Campana *, Ricotta Romana *	Greece	*Anthotyros, Myzithra, Manouri, Xynomyzithra, Urda,* *Xinomyzithra Kritis* *
Norway	Mysost Gjetost, Brunost	Turkey	*Lor Peyniri, Tire Çamur Peyniri*
Sweden	Mesost, Messmör	Cyprus	*Anari*
Denmark	Myseost	Croatia	*Skuta*
France	Brocciu corse *	Serbia	*Skuta, Urda*
Romania	Urda	Mexico	*Requesón*
Portugal	Requeijão, Requeijão da Beira Baixa *, Requeijão Serra da Estrela *	Iceland	*Mysuostur*
Spain	Requesón		

* European Commission, eAmbrosia Database, accessed 2025 [132]. Source: based on [86,133,134,135].

**Table 4 foods-14-04258-t004:** Advantages and disadvantages of whey valorisation for the production of bioethanol, biogas, and biohydrogen.

Whey Valorisation	Yields and Operating Conditions	Techno-Economic Assessment (TEA)/Life-Cycle Assessments (LCA) Indicator
Bioethanol	Ethanol fermentations with *Kluyveromyces marxianus* are often carried out at 30–40 °C with pH ~4.5 (or without strict pH control). Using *K. marxianus* yields of ~0.50 g ethanol per g lactose have been reported in batch fermentation at 30–40 °C and acidic pH (~4.5), with productivity up to 2.5–4.5 g/L·h in continuous alginate-immobilized systems. It was found that when the initial whey was 18.8 g/L, fermentation with K. marxianus for 12 h reduced the COD content by 82.28%.	The TEA value of a scaled-up whey-to-ethanol facility (developed in a simulation study) showed a strong dependence on the “gate fee” for whey. When whey is treated as a waste stream (negative cost or credit for the processor), the economics improve significantly, giving a minimum ethanol selling price of ~1.43 €/kg for Y_E/L = 0.45 g/g.
Biogas (methane)	Anaerobic digestion for methanogenesis typically runs mesophilic (e.g., ~35 °C in UASB reactors) or thermophilic for higher rates. In single-stage anaerobic digestion of whey under thermophilic conditions, methane yields of around 314 L CH_4_ per kg COD added were observed. Under mesophilic UASB digestion, other studies reported ~120 L CH_4_/kg COD.	In life-cycle assessments (LCA) and economic modeling of two-stage anaerobic digestion (for H_2_ + CH_4_), studies show that on-site systems yield favorable greenhouse gas (GHG) savings compared to conventional disposal, making them attractive for medium-scale dairies.
Biohydrogen (H_2_)	Anaerobic fermentation of whey/lactose has achieved 2.8–3.6 mol H_2_ per mol lactose under buffered conditions (COD ~9–15 g/L). Theoretically, higher yields (up to 8 mol H_2_/mol lactose) are possible if acetate is the only end-product For dark hydrogen fermentation, an initial COD of around 9–15 g COD/L, with a buffering system (e.g., MES buffer) to maintain pH ~5–5.5, yields the best H_2_.	Energetic efficiency (energy consumption) for processes like aqueous-phase reforming (APR) of dairy wastewater has also been studied: for example, producing H_2_ in a reforming plant gave LCA footprints of 2.57 kg CO_2_-eq per kg H_2_, and an energy cost (minimum selling price) of ~USD 7.00/kg H_2_ in one scenario.

Source: own study based on literature [67,188,189,190,191].

**Table 5 foods-14-04258-t005:** SWOT analysis of whey valorisation.

Strengths	Weaknesses
Whey is a rich source of vitamins, minerals, proteins, and other nutritional values. Therefore, it can be used to develop new, nutritional products. The valorization of whey is the achievement of the UN Sustainable Development Goals, such as Goal 6. Clean Water and Sanitation, Goal 12. Responsible consumption, and Goal 13. Climate Action. This is in line with the principles of a circular economy (Zero waste).	Whey recovery and valorisation from small dairies can be difficult and too expensive. Valorisation processes are not always available or affordable for small-scale producers who do not have the finances, resources, or infrastructure. Some types of whey are difficult to valorize (e.g., acid whey cannot be valorised through membrane filtration).
**Opportunities**	**Threats**
The development of modern processing methods (technologies) may be more effective in the valorisation of whey. Waste whey from small dairies can be sold for further processing. Consumer interest in new products, such as probiotics, whey beverages, and other functional products, may increase the dairy industry’s interest in whey valorisation.	Excessive production of dairy products may result in increased environmental pollution, despite the dairy industry’s growing interest in whey valorisation. Failure to valorize leads to significant economic, nutritional, and environmental losses. Whey processing also creates wastewater and environmental pollution. Due to the fragmented producer structure in the dairy industry, there is a lack of comprehensive, documented data on discharges to wastewater or soil, as well as data on whey valorization. As a result, this leads to a lack of comprehensive solutions for whole industry.

Own study.

## Data Availability

No new data were created or analyzed in this study. Data sharing is not applicable to this article.

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
