# Peer review of "Whey—A Valuable Technological Resource for the Production of New Functional Products with Added Health-Promoting Properties"

_foods, 2025, doi:10.3390/foods14244258_

Round 1

Reviewer 1 Report

Comments and Suggestions for Authors

This comprehensive review effectively synthesizes current knowledge on whey valorization, emphasizing its transformation from an environmental burden to a resource for functional foods and sustainable technologies. The manuscript aligns with global sustainability goals and addresses a timely topic in food science. While well-structured and informative, it requires minor revisions to enhance clarity, methodological transparency, and critical analysis.

Major Comments

Methodology Clarity (Section 2):

The literature search strategy (PubMed/Web of Science/Scopus, 2015–2025) lacks specificity. Define inclusion/exclusion criteria (e.g., study types, languages) and detail the screening process (e.g., PRISMA flow diagram suggested in Fig. 1 but not provided).

Clarify how data synthesis was performed (e.g., thematic analysis, meta-analysis).

Structural Balance:

Sections 3.1 (global whey production) and 3.7 (technological problems) are robust, but 3.3 (microbiology) is overly detailed for a technological review. Condense microbiological mechanisms and focus on implications for valorization (e.g., fermentation efficiency, safety).

Section 3.6.3 (functional beverages) could be streamlined; some paragraphs repeat fermentation benefits.

Critical Analysis Gaps:

While Table 3 summarizes whey applications, the text lacks comparative critique of methods (e.g., cost-effectiveness of membrane tech vs. enzymatic hydrolysis). Discuss trade-offs (e.g., energy use in novel processing).

Address contradictory data: Claims that "only 25% of whey is used for human consumption" (p. 4) conflict with later statements (e.g., "70–90% processed in industrialized countries," p. 18). Reconcile discrepancies.

Environmental Impact Depth:

Expand on circular economy frameworks (Section 1). Quantify environmental benefits of valorization (e.g., reduced BOD/COD, carbon footprint) beyond Table 2. Link to UN Sustainable Development Goals explicitly.

Author Response

Dear Reviewer 1, 

Thank you for reviewing our manuscript. Revisions were made according to the reviewers’ comments. Changes in the manuscript are indicated in Track revisions. The text was proofread by a Proofreader from Smaller Earth, Poland. We attached the certificate.

We hope that the revised manuscript will find your acceptance for publication. Thank you for your patience and help.

with best regards

authors

Reviewer 2 Report

Comments and Suggestions for Authors

This manuscript, titled “Whey - a valuable technological resource for the production of new functional products with added pro-health properties” presents an interesting review paper.  The manuscript provides a clear overview of whey’s transformation from a by-product into a strategic raw material. The topic is timely and highly relevant to sustainable food production and circular bioeconomy approaches. Despite its massive content, the paper reads descriptively rather than critically. Facts are often summarized without mentioning methods, limits, or comparisons. Well-prepared figures and tables require clearer legends and reference.

-Firstly, the resolution of figure 1 is not good. Please increase the quality of the figure.

-Line 151, 158; Rephrase the sentences to improve clarity.

-The technologies are given in the order that they were developed in Section 3.5 (Novel whey processing techniques), however they are not compared with one another in terms of their effectiveness, cost, sustainability, or product yield. Please add a comparative discussion or a summary table comparing these approaches

-The figures (for example; Figure 1 – Timeline of whey management) are informative, but legends lack explanations for the abbreviations and symbols such as WPC, WPI, DWP.

-Captions should state, "Adapted from Ref. X with permission," or that figures were made by the authors. On the other hand, authors should provide copyright permissions where necessary.

-Table 3 provides whey protein preparations and might use a column for particular results or performance indicators (for example; bioactivity tested, protein recovery %).

-Table 2 shows wastewater data without n or standard deviations. Additional dataset size or range information would improve transparency.

-Researchers should indicate all abbreviations upon first mention (BOD, COD, WPC, WPI, etc.).

-The section titled Conclusions is quite strong, but it would be beneficial to include a Future Perspectives subparagraph that summarizes the present gaps and directions.

-Line 373–403: Improve sentence transitions; currently, the list of methods reads like notes rather than a narrative.

Author Response

Dear Reviewer, 

Thank you for reviewing our manuscript. Revisions were made according to the reviewers’ comments. Changes in the manuscript are indicated in Track revisions. The text was proofread by a Proofreader from Smaller Earth, Poland. We attached the certificate.

We hope that the revised manuscript will find your acceptance for publication. Thank you for your patience and help.

with best regards

authors

Reviewer 3 Report

Comments and Suggestions for Authors

This is a comprehensive and well-structured review that covers technological, environmental, and nutritional aspects of whey valorization. The topic is timely and aligns well with current sustainability goals. The manuscript is generally well-written, but several sections require clarification, consistency, and more precise language.

1.Title: "pro-health" should be "health-promoting" for better readability.

2.Lines 22-22: The statement "Data from 2015 to 2025 were collected..." is misleading, as the year 2025 is still ongoing. Suggesting to rephrase to "Data from 2015 to the present (2025) were collected..." or specify the exact date of the literature search.

3.Line 209-210: "popcorn whey" is not well explained. Consider a brief clarification or synonym.

4.Conclusion: The conclusion is strong but could be more forward-looking. Suggest adding 1-2 sentences on future research priorities (e.g., cost-effective membrane technologies, standardization of whey composition).

5.Figure 1: There are two Figure 1 in the manuscript. Please correct.

6.Table 1: Several entries have "n/d" (no data) without explanation, which weakens the table's utility. Suggesting to add a footnote explaining why data are unavailable or estimate based on available references.

7.Table 3: Some entries in the "Technological application" column are overly vague (e.g., "Functional foods"). Suggesting to specify examples or categories (e.g., "sports drinks, infant formula") to enhance practical relevance.

Author Response

(The authors gave the same response as above.)

Reviewer 4 Report

Comments and Suggestions for Authors

foods-3986404-peer-review-v1

  1. The authors provide a good overview of the topic. However, the novelty of the work (and how it is filling the current gap) is missing. Has any similar study been published before? What difference does your work make?
  2. The connection between the "Materials and Methods" section (Section 2) and the "Results" (Section 3) could be clearer. The search strategy is mentioned, but the synthesis process (e.g., how studies were selected for inclusion, criteria for emphasizing certain technologies over others) is not described. A PRISMA-style flow diagram (as hinted in Figure 1 of the manuscript, which is actually an article search scheme) would be highly beneficial for transparency. Please ensure the caption for this figure is correctly placed and referenced.
  3. The main findings of the application of whey in the specific products should be extracted in a table form.
  4. The conclusion effectively summarizes the content, but could be more forward-looking. It should more strongly emphasize the key research gaps identified in the review and propose specific, promising directions for future research.
  5. Some sections feel like a list of facts rather than a flowing narrative. For example, Section 3.6.4 ("Other usage of whey") jumps from biofuels to edible films to iron fortification without strong transitional sentences linking these diverse topics. Consider restructuring this section into clearer sub-sections or improving the narrative to guide the reader through the logical connections between these applications.
  6. The review is largely descriptive. It would significantly benefit from a more critical discussion that compares and contrasts different valorization pathways. For instance, in Sections 3.3 (Microbiological Overview) and 3.6 (Utilization), the advantages and disadvantages of various fermentation approaches (lactic vs. alcoholic) or different beverage formulations could be more explicitly discussed.
  7. The unpublished article should be removed from the reference list. Ref. 38

Author Response

(The authors gave the same response as above.)

Reviewer 5 Report

Comments and Suggestions for Authors

It has been a challenging task to review the article entitled “Whey - a valuable technological resource for the production of new functional products with added pro-health properties” by Ewa Czarniecka-Skubina et al.
The manuscript presents a comprehensive and informative review of whey valorisation, covering production scale, environmental impacts (BOD/COD), technological advances (membrane, enzymatic, and non-enzymatic processes), and applications in food and bioproducts. The topic is current and relevant within the context of the circular economy and innovation in functional foods. The work gathers a wide range of references and practical examples; however, its scientific and editorial quality could benefit from improvements in clarity and conciseness of writing, terminological consistency, coherence of data and units, bibliographic support for numerical values, and harmonisation of tables, figures, and regulatory aspects.
The following are detailed and actionable comments, organised by section, with specific proposals to enhance methodological rigour and align the manuscript with publication standards.

Abstract

L18–19 – “Over time, whey evolved…” shows temporal inconsistency; the abstract should use the present or present perfect tense.
L20–21 – “Data from 2015 to 2025 were collected…” The future year (2025) without a month indication is incorrect, as it implies the inclusion of unavailable data. Please revise.
L22–24 – The list of technologies (“ultrafiltration, nanofiltration, thermosonication, and fermentation…”) is too long and disrupts sentence rhythm. Consider summarising or dividing into two sentences, briefly indicating technological relevance.
L27–28 – “200 million tonnes of whey produced annually” – please insert an updated reference (FAO, IDF, or recent 2023–2024 publication).

Introduction

L35–37 – The phrase “formed in the natural fermentation during the manufacture of cheese, casein, or similar products…” is unclear and redundant. Suggested revision: “Whey is the liquid by-product remaining after milk coagulation during cheese or casein manufacture.”
L39–40 – Revise to “The pH, determined by the milk coagulation method, defines the main types of whey.”
L41–47 – The description of whey types is correct but repetitive (“which is produced”, “is achieved”, “are used”). Suggested simplification: “Three main types of whey can be distinguished: (i) sweet whey (pH 5.9–6.3), (ii) acid whey (pH 4.3–4.6), and (iii) casein whey (pH 4.6–4.7).”
L48–50 – “vitamin C, B vitamins, B2, and B6” is redundant. Replace with “vitamin C, riboflavin (B2), and pyridoxine (B6).”
L53–54 – The exact percentages (“95% albumin, 95% globulin…”) should be verified; use “approximately” and add a supporting reference.
L56–63 – The paragraph on environmental pollution is too long and descriptive. Divide into two sentences for clarity and include BOD/COD values to quantify the environmental impact of whey. These parameters are key indicators of effluent organic load. BOD values of 30–50 g/L and COD values of 60–80 g/L reflect its high pollution potential, confirming the need for pre-treatment before discharge.
L62–63 – “causes the destruction of aquatic ecosystems through the loss of dissolved oxygen” is accurate but requires a reference.
L73–81 – The paragraph describing objectives is redundant (“assess”, “discuss”, “identify”, “presented”). Suggested condensation: “This review aims to assess the potential of whey valorization and to discuss current technological, environmental, and nutritional strategies for its utilization in the context of circular economy and sustainable development.”
L80–81 – In “which align with the global sustainable development goals.” please add a reference (e.g., UN SDGs 12 and 13).

Materials and Methods

L83–84 – “All data presented in this review were summarized from references…” is vague. Specify the review type (e.g., narrative, systematic, or scoping review).
L85–86 – “The references were searched in PubMed, Web of Science, and Scopus…” Authors should specify the exact date of the search (day/month/year) and the inclusion/exclusion criteria (e.g., language, document type, duplication handling, and thematic relevance).
L86–88 – Improve the search string for consistency, e.g.: “The following search string was used: (whey) AND (‘technological process’ OR ‘processing technology’) AND (‘utilization’ OR ‘valorization’).”
L88–89 – “search scope was limited to the years 2015–2025.” The use of 2025 is incorrect, as the year is not complete. Indicate the exact cut-off date.
L90 – Figure 1 – The caption “Article search scheme” is too generic. Revise to “Figure 1. Flow diagram of literature search and selection process.”
Figure 1 – The flowchart is well structured but contains minor inconsistencies:

“Web of Science & Scopus n=978” should also include PubMed,

“Record Removed (repetition, out of topic)” → “Duplicates and non-relevant records removed.”

“Record Included n=164” → clarify inclusion criteria (e.g., research articles or reviews only).
Recommend aligning the scheme with a simplified PRISMA flow diagram for methodological transparency.

L97–99 – “1 kg cheese / 10 L milk → 9 L whey” – indicate variation by cheese type (hard vs soft) and provide reference.
L106–111 – Check coherence of percentages (“50–60% valorized; 42–50% untreated; 25% for human consumption”). Clarify definitions of “valorized”, “low-value uses”, and “human consumption.”
L117–122 – This paragraph on methodological limitations is useful; add 1–2 sentences explaining how the current review addressed these gaps (e.g., data triangulation, exclusion of duplicates).

Environmental and Process Data

L126–137 – Numerical inconsistencies: BOD is first said to be 30–50 g/L, then “sweet whey 40–102 g/L,” which contradicts earlier values. Similarly, COD 60–80 g/L vs. “sweet 40–102 g/L; acid 52–62 g/L.” Revise these intervals (BOD and COD should not share identical ranges). Also, acid whey is described as “lower in lactose,” yet reported with equal or higher COD/BOD—please verify.
L173–179 – Table 2 – BODâ‚… and COD values appear swapped or incorrect, possibly due to transcription errors (“BODâ‚… = 5 312 mg/dm³”, “COD = 20 559 mg/dm³”, then “COD = 5 312 mg/dm³” under recovery). Verify units and source values [10, 38]; COD should exceed BODâ‚… (ratio ≈ 2:1). Replace “mg/dm³” with “mg/L” and expand the legend to explain environmental significance, confirming COD > BODâ‚….
L187 – Update to “Figure 2. Evolution of whey processing (1960–2025)”; standardize icons and years; add WPI (2000) and WPH (2010) consistently.
L237–257 – Technologies are listed without parameters. For 2–3 key techniques (UF/NF, SMB chromatography, membrane adsorption), provide metrics (recovery %, purity %, productivity, energy consumption kWh/kg).

Microbiology and Functional Aspects

L258–338 – Reorganize this subsection into: (i) typical microbiota (LAB/yeasts); (ii) LAB–yeast interactions; (iii) technological implications; (iv) risks and safety. Ensure genus/species names are italicised and nomenclature updated (e.g., Kluyveromyces marxianus, Lactococcus lactis, Lactobacillus helveticus, Streptococcus thermophilus).
L322–337 – “Resistance of S. aureus enterotoxins to 100–121 °C/3 min” – specify conditions (pH, matrix, water activity) and cite primary sources; avoid overgeneralization (“may not be sufficient in some matrices/conditions”).

Applications

L423 – Table 4 – Suggested corrections: Remove duplicate “Manouri.”; Correct spelling “Tire Ç amur Peyniri” → “Tire Çamur Peyniri.”; Normalize cheese names in the original language and in italics.; Clearly separate or mark PDO/PGI cheeses, with updated registration year (consult EU eAmbrosia Database); Replace “EU GI Register, 2003” with “European Commission, eAmbrosia Database, accessed 2025.”

Clarify inclusion criteria in the caption, e.g.: “Examples of whey cheeses traditionally produced from milk whey, including those recognized under PDO/PGI status in the European Union.”

L505–509 – Revise “no clear regulations” and reference the EFSA/Regulation (EC) 1924/2006 framework and FAO/WHO probiotic guidelines.
L538–561 – Add typical yields (g ethanol/L; m³ CHâ‚„/t COD; Hâ‚‚ mol/mol lactose), operating conditions, and TEA/LCA indicators (kWh/kg, CAPEX/OPEX, payback).
L543–546 – The expression “6.4 mL ethanol / 250 mL whey” is not informative; convert to g/L or % v/v and theoretical yield %. Include density/temperature.
L562–571 and L603–612 – The blocks on “whey cream/iron chelation” are duplicated; please merge into one subsection (nutrition/iron chelates) with consolidated data and references; keep “whey cream” under dairy applications, not packaging.

Technological Problems

L613–685 – Reorganise section 3.7 for clarity:  (i) raw material variability; (ii) process challenges (acid vs sweet whey); (iii) microbiological safety/quality; (iv) economic and scale aspects; (v) environmental/regulatory issues.
This structure will avoid redundancies, improve logic, and highlight key technological barriers.
L637–638 – “whey cannot be valorised through membrane filtration due to lactose crystallization” is too absolute. Acid whey can be treated by NF/RO/diafiltration/electrodialysis for demineralisation or lactose recovery; crystallisation is mainly a drying/storage issue. Revise and cite.

Conclusions

L686 – Structure conclusions in three bullets: (i) scale and impact (production/valorization, BOD/COD); (ii) technological advances (membrane, enzymatic, non-thermal) with 1–2 quantitative examples; (iii) remaining gaps and priorities (global data, standardization, TEA/LCA, probiotic/packaging regulations).
L694–701 – Move market data to Introduction or Discussion, with the commercial source cited (year/methodology). Conclusions should summarize findings, not introduce new data.
L699–701 – Remove references to “immune-boosting/cancer patients” unless supported by strong clinical evidence; otherwise, soften (“are being explored…”).
L725 – Add a short paragraph on methodological limitations (fragmented data, estimations) and research agenda (acid whey, bitter peptide removal, film scalability, water reuse, circularity metrics).

Comments on the Quality of English Language

The English could be improved to more clearly express the research.

Author Response

(The authors gave the same response as above.)

Round 2

Reviewer 4 Report

Comments and Suggestions for Authors

All questions have been addressed in the revised manuscript.

Reviewer 5 Report

Comments and Suggestions for Authors

The manuscript review is excellent. In my opinion, the manuscript should be approved for publication. However, I noticed that the line count increases from 143 to 150. I recommend that the authors review this discrepancy.